# Intact HIV DNA decays in children with and without complete viral load suppression

**Daniel B Reeves** [1,2]*, **Morgan Litchford**[3], **Carolyn S Fish**[3], **Anna Farrell-Sherman**[1], **Makayla Poindexter**[1], **Nashwa Ahmed**[1], **Noah A J Cassidy**[3], **Jillian Neary**[2], **Dalton Wamalwa**[2,4], **Agnes Langat**[4], **Daisy Chebet**[4], **Hellen Moraa**[4], **Annukka A R Antar**[5], **Jennifer Slyker**[2], **Sarah Benki-Nugent**[2], **Lillian B Cohn**[1], **Joshua T Schiffer**[1,6], **Julie Overbaugh**[3], **Grace John-Stewart**[2,6], **Dara A Lehman**[2,3]

**1** Vaccine and Infectious Diseases, Fred Hutchinson Cancer Center, Seattle, Washington, United States of America, **2** Global Health, University of Washington, Seattle, Washington, United States of America, **3** Human Biology, Fred Hutchinson Cancer Center, Seattle, Washington, United States of America, **4** Department of Pediatrics and Child Health, University of Nairobi, Nairobi, Kenya, **5** Department of Medicine, Johns Hopkins University, Baltimore, Maryland, United States of America, **6** Department of Medicine, University of Washington, Seattle, Washington, United States of America

* dreeves@fredhutch.org

## Abstract

To inform cure in children living with HIV (CWH), we elucidated the dynamics and mechanisms underlying HIV persistence during antiretroviral therapy (ART). In 120 Kenyan CWH who initiated ART between 1-12 months of age, 55 had durable viral load suppression, and 65 experienced ART interruptions. We measured plasma HIV RNA levels, CD4+ T cell count, and levels of intact and defective HIV DNA proviruses via the cross-subtype intact proviral DNA assay (CS-IPDA). By modeling data from the durably suppressed subset, we found that during early ART (year 0-1 on ART), plasma RNA levels decayed rapidly and biphasically and intact and defective HIV DNA decayed with mean 3 and 9 month half-lives, respectively. After viral suppression was achieved (years 1-8 on ART), intact HIV DNA decay slowed to a mean 22 month half-life, whilst defective HIV DNA no longer decayed. In five CWH, we found individual CD4+ TCRβ clones wax and wane, but average kinetics resembled those of defective DNA and CD4 count, suggesting that differential decay of intact HIV DNA arises from selective pressures overlaying normal CD4+ T cell kinetics. Finally, by modeling HIV RNA and DNA in CWH with treatment interruptions, we linked temporary viremia to transient rises in HIV DNA, but long-term intact reservoirs were not strongly influenced, suggesting brief treatment interruptions may not significantly increase HIV reservoirs in children.

## Author summary

HIV remains a difficult prognosis for children because it is deadly if untreated and therefore requires a lifetime of antiretroviral therapy (ART). Here we analyzed data from a cohort of children in Nairobi Kenya who acquired HIV and determined that the reservoir of latently infected cells that persist during ART, and are the barrier preventing cure, decays consistently but slowly over 10 years of ART. We provide detailed estimates

**Data availability statement:** All data and code used to perform the analyses and modeling in this work is freely available from https://github.com/dbrvs/OPHmodeling

**Funding:** Funding was provided by the NIH R01s HD023412 & HD094718 to DAL and GJS, K25 AI155224 and R01 AI186721-01 to DBR. LBC is supported by UM1AI164565. DBR is thankful for the support of the University of Washington/Fred Hutch Center for AIDS Research (CFAR, P30 AI027757) through a New Investigator Award and a Supplement that initiated and funded this project. The funders had no role in study design, data collection and analysis, decision to publish, or preparation of the manuscript.

**Competing interests:** The authors have declared that no competing interests exist.

of HIV reservoir decay rates for the subset of children that suppressed their viral loads and relate decay to immunological factors (such as CD4+ T cell levels and clonality). Using mathematical modeling, we also showed that brief treatment interruptions do not increase long-term reservoir levels. Our results could inform cures targeting immunological factors and the handling of treatment interruptions, whether planned in clinical trials or incidental in the real world.

## Introduction

Children are an especially relevant population for HIV cure both because they face a lifetime prognosis of ART and because known exposures from mothers enables earlier treatment and better acquisition time estimates than typical for adults [1,2]. Though antiretroviral therapy (ART) vastly improves survival of perinatally infected children living with HIV (CWH) [2], mortality and adverse outcomes remain more common in CWH on ART than in children without HIV [3–5]. Moreover, suppressive ART is not curative. CWH must continue ART due to a reservoir of latently infected cells harboring HIV DNA proviruses [6,7] which can seed rebound viremia if ART is stopped [8,9]. Very-early ART for confirmed *in utero* HIV acquisition powerfully restricts reservoir size [10] and smaller reservoirs are associated with longer times to viral load rebound if ART is stopped [11–14]. This was exemplified by early treated CWH like the highly publicized 'Mississippi baby,' who experienced ART-free remission for almost two years [1,15,16]. Yet, the eventual viral rebound of this exceptional child highlights why we still need a better understanding of the mechanistic underpinnings of pediatric reservoir persistence, including immune correlates of reduced reservoir size or decay.

Several mechanistic insights into reservoirs have been revealed by specifically assaying intact proviruses. Intact proviral DNA assays (IPDA) [17–21] quantify defective HIV proviruses and those that are intact at two or more conserved locations, providing an approximation of the rebound competent reservoir [22]. IPDA studies in adults have shown intact HIV proviruses decay faster than defective HIV proviruses [17,23–26]. Pediatric studies also showed this "differential decay" [10,27] but have mostly addressed the first 2 years of ART and/or have been limited geographically to US participants with predominantly subtype B HIV-1. Thus, previous studies might not be representative of long term kinetics in subtype B nor be representative of kinetics in children in global HIV hotspots such as sub-Saharan Africa where other subtypes (e.g., A, C, and D) predominate.

Proliferation of latently infected cells is a key mechanism driving HIV reservoir persistence [23,28–31]. In adults, we found that HIV proviral sequences and memory CD4+ T cell receptor β (TCRβ) sequences both waxed and waned over time [32]. Moreover, modeling longitudinal data suggested that many drivers of HIV persistence could be the same as those maintaining memory CD4 T cells [33]. Meanwhile, certain immune markers related to cell proliferation, inflammation, and signaling have been correlated with intact reservoirs in adults, leading us to hypothesize that interleukins and/or Granzyme B, which mediate proliferation and/or apoptosis of memory CD4+ T cells could influence HIV reservoir kinetics [34,35].

To address several of these issues, we used our cross-subtype IPDA (CS-IPDA) assay [18,19,36] to study the Optimizing Pediatric HIV (OPH) cohort, an early-treated cohort of 120 Kenyan children that was followed for 10 years [8,37]. This powerful cohort has illuminated that viral suppression is harder for younger children than older children [38] and that ART adherence remains a key issue for CWH [38,39], with adolescents in particular often struggling to maintain durable viral suppression [40]. For this group, immunosuppression,

first-line ART regimens, and cytomegalovirus and Epstein Barr acquisition age and viremia associated with HIV DNA seeding and maintenance [41]. A randomized trial in this cohort (NCT00428116) assessed analytical treatment interruption (ATI) in 21 children at year 2 of ART.

Our goal in the present work was to rigorously characterize intact and defective HIV DNA kinetics in a cohort with representative characteristics of the global pediatric HIV population. We quantitated decay and expansion rates for HIV RNA and DNA and CD4+ T cells. We also explored factors contributing to reservoir decay including immune markers and CD4+ T cell sequencing. Finally, given the challenges for ART adherence and virologic suppression in this population, we developed a model that allowed us to include participants with periods of viremia and understand how treatment interruptions in CWH influence HIV reservoirs, which is key to informing the ethics of performing ATI trials, the gold standard for assessing HIV cure [42,43].

## Results

### The OPH cohort: early ART in Kenyan children living with HIV

The Optimizing Pediatric HIV (OPH) cohort is comprised of 120 children from Nairobi, Kenya who acquired HIV through vertical transmission from their mother and initiated ART early after birth (**Fig 1A**); median age at ART start was 3 months (range 1-17 months) with even male/female distribution (**Fig 1B**). Follow up continued for up to 10 years after ART initiation. Based on HIV pol sequencing for N=106 (88%) of the 120 children included here, HIV-1 subtypes were split into 68 (64%) subtype A, 10 (9%) subtype C, 21 (20%) subtype D, and 7 (7%) recombinant (**Fig 1C**).

### Longitudinal measurements of plasma viral load, CD4 count, intact and defective HIV proviruses, immune markers and TCR sequencing during 10 years of ART

To determine the dynamics of HIV reservoirs and related viral and cellular variables in OPH participants, CD4+ T cell counts (cells/μL), and plasma viral loads (log10 HIV RNA copies/mL) were obtained every 3-6 months, resulting in median of 19 (range 1-32) viral loads and 11 (range 1-24) CD4 counts per child over a 10 year period (**Fig 1D**). There were 95 participants who had viral load measurements at 12 months, 75 at 40 months, and 57 at 96 months – decrease over time was due to death and loss of follow-up.

The cross-subtype intact proviral DNA assay (CS-IPDA) was used to quantify intact and defective HIV provirus levels. We measured HIV DNA in copies per million T cells given the smaller numbers of cells available from the pediatric samples. Samples were available for CS-IPDA measurements from 83 participants from time points at ART initiation (or just before) and months 6, 12, 24, 40, 60, and 96 on ART (**Figs 1D** and **S1**). There were a median of 5 (range 1-10) longitudinal CS-IPDA measurements per child over the whole study.

Additionally, four immune markers (IL-2, IL-7, IL-15, GzB) were measured in plasma from 30 participants at or close to CS-IPDA time points, and non-naive CD4+ T cell receptor β-chain repertoire sequencing was performed on 5 participants at roughly month 40 and 96 on ART (**Fig 1D** and **Methods**).

### Validating the CS-IPDA assay for OPH samples

CS-IPDA is a multiplexed droplet-digital PCR assay that targets 3 HIV-genomic regions (*gag, pol* and *env)* [18,19]. Our primers were designed based on the sequences available in the Los Alamos National Lab (LANL) database in 2019-2020 (https://www.hiv.lanl.gov/content/

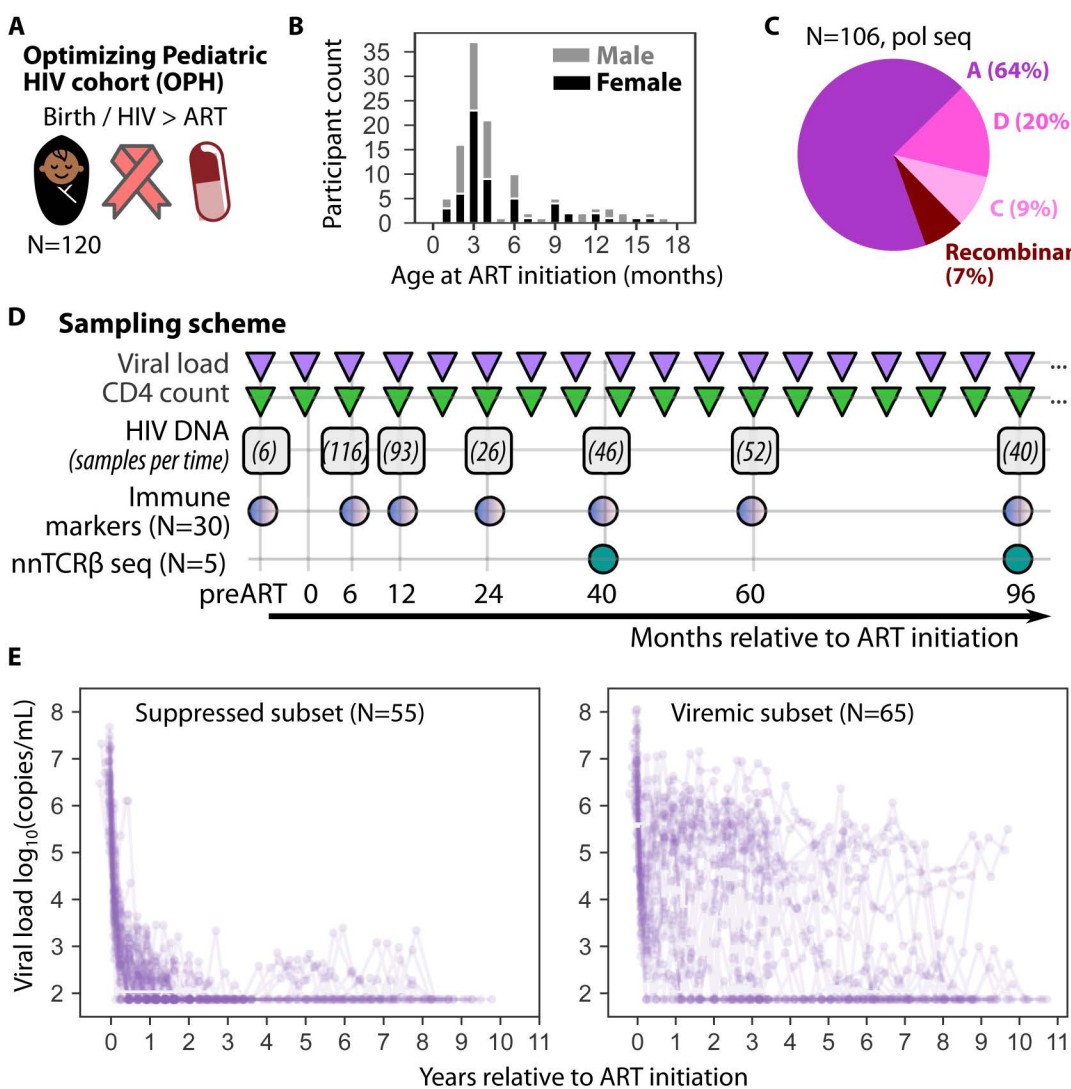

**Fig 1. OPH cohort data and study design.** *A) Overall cohort comprises 120 Kenyan children who acquired HIV near birth and received early ART. B) HIV-1 subtype from 106 participants sequenced via pol. C) Histogram of ART initiation age colored to show balance of male and female children. D) Longitudinal measurement timing is noted for viral load (log10 HIV RNA copies/ mL) and CD4 count (cells/μL) as approximately every 6 months for all participants. Intact and defective HIV DNA using the CS-IPDA assay were measured at 7 time points and number of participants sampled at each time point are noted. Immune markers (N=30 children) were measured at the HIV DNA time points. Non-naïve TCRβ sequencing (N=5 children) was performed at two long-term ART time points. E) Viral loads for the suppressed and viremic subsets of OPH participants, stratified based on whether viral load dropped below 1000 copies/mL within the first year of ART and stayed below 3000 copies/mL thereafter.*

sequence/HIV/mainpage.html), which included Kenyan samples from the time the participants in the OPH cohort acquired HIV (2007-2010) [8].

Although we do not have full length HIV sequences to verify intact versus defective proviruses as classified by our CS-IPDA, we determined the proportion of samples for which our CS-IPDA failed to detect any signal for any of the 3 targets (similar to [44]). Complete lack of signal detection for a target region could either indicate assay failure due to polymorphisms in the primers and/or probe binding sites (potentially intact sequences that are called defective) or could indicate true missing genomic regions due to deletions (true defectives that are called

defective). Only 10 of 356 (3%) samples with CS-IPDA measurements had no signal from a target. Importantly, every CS-IPDA target was detected in at least one sample time point for all individuals tested here. This is a very low failure rate compared to other published IPDAs [44] and suggests our CS-IPDA effectively works across the HIV subtypes in this cohort.

## Splitting suppressed and viremic subsets of participants in OPH

Given the challenges of viral suppression in CWH, we defined a "suppressed subset" that achieved HIV viral loads below 1000 copies/mL by 1 year after ART initiation and then remained below 3000 copies/mL thereafter. Overall, 55 CWH, representative of the OPH cohort age and sex distribution, met criteria for the suppressed subset with 65 participants comprising the viremic subset (viral loads in **Fig 1E**).

Though 3000 copies/mL is above standard pediatric definitions of durable suppression, this higher level enabled us to include several more participants with key long-term HIV provirus measurements. To ensure this definition was not unreasonable, we performed additional analyses. First, as would be expected, we showed intact and defective HIV provirus levels were higher in the viremic vs suppressed participants during and after the first year of ART (S2 Fig, all p<1e-5). Second, examining time points after the first year of ART in the suppressed subset, we found intact and defective HIV provirus levels were not significantly different between time points with low but detectable viral loads (below 3000 copies/mL) vs. undetectable viral loads (S3 Fig). Third, we examined the relationship between HIV RNA viral load area under the curve (AUC) and final intact HIV DNA level (S4 Fig) and found no strong statistical relationship (Spearman p=0.3), suggesting that occasional viremia does not strongly alter long-term reservoir size. Finally, when we estimated decay rates from these data, we also performed sensitivity analyses with more stringent suppression criteria: HIV viral loads below 1000 or 300 copies per mL after initial suppression and show similar results despite the smaller sample sizes (see below).

## Population dynamics of viral loads, CD4 counts, and intact and defective HIV proviruses in children who achieve viral load suppression

In the suppressed subset of 55 CWH as defined above, we applied population nonlinear mixed effects (pNLME) modeling to estimate kinetics of viral loads, CD4 counts, and intact and defective HIV proviruses. To avoid biases from decreasing observation numbers over study years, we estimated rates during and after the first year of ART separately (**Fig 2A**).

HIV viral loads decayed with a mean half-life of 6 days (95% CI: 5–9 days) for the first 2 months of ART, and thereafter a mean half-life of 2.7 months (95% CI: 2–4.6 months) until suppression was achieved at the end of the first year of ART (note suppression is generally slower than typical for adults). In the first year of ART, intact and defective HIV proviruses decayed differentially with mean half-lives of 2.7 months (95% CI: 2–4.1 months) and 8.6 months (95% CI: 5.4–20.7 months), respectively. After 1 year of ART, intact proviruses continued to decline, but more slowly, with a mean half-life of 22 months (95% CI: 10.5–∞ months), whereas defective HIV proviruses stabilized or increased extremely slowly with a doubling time of 101 months (95% CI: 54–845 months) (**Fig 2B**).

Concurrently, CD4 counts rose with a mean doubling time of 15 months (95% CI: 11.6–21.1 months) in the first year on ART and thereafter declined with a mean half-life of 7.8 years (95% CI: 6–10.9 years). These CD4 kinetics were similar to three published studies of infants without HIV [45–47] (S5 Fig).

All population decay rates, standard errors, and model error (% residual squared error) are reported in S1 Table and kinetics are visually summarized (relative to baseline level) in **Fig 2C**.

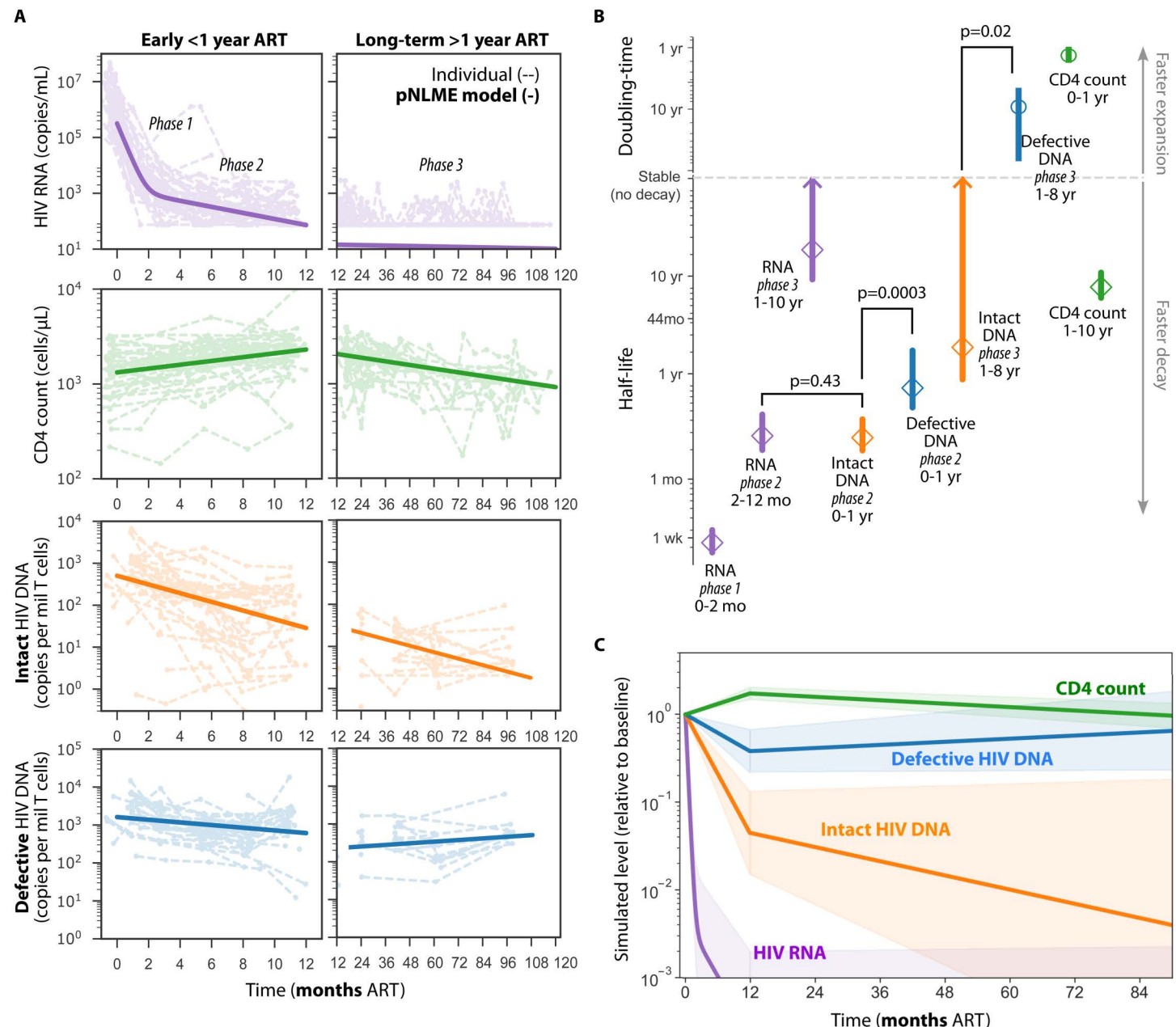

**Fig 2. Modeled kinetics in suppressed subset.** *A) For viral load, CD4 count, and intact and defective HIV DNA, exponential population nonlinear mixed effects modeling (pNLME) fit to data separated by early (<1 year, left) and long-term (>1 year, right) ART. Note months vs years in x-axes. Dashed lines indicate individual trajectories with dots at observed time points. Bold lines indicate pNLME population mean. B) Estimated population means (dots) and 95% confidence intervals (CI, lines) for each data type. Phases were defined as the first 0-2 months, 2-12 months for HIV RNA and 0-1 year for HIV DNA, and 1-8 years on ART. Axes are split by no decay (or very stable kinetics) such that half-lives and doubling times can be presented. Arrows on CI indicate overlap with no decay. Comparisons indicate 1-sided Z-test. C) Simulation summary of mean (line) and 95% CI (shading) kinetics relative to baseline level for each data type.*

## Multiphasic differential decay of intact and defective HIV DNA in children

We observed three phases of decay for HIV RNA, and two phases for HIV DNA. However, others have observed that (in adults) HIV DNA decay also could have two phases during 0-1 year on ART (as for RNA [48]). Indeed, the second phase decay of HIV RNA (2-12 months

on ART) was similar to the decay rate of HIV DNA in the first year of ART (p=0.43, **Fig 2B**). Thus we speculated that there could be an additional first phase of HIV DNA decay that may be observed with tighter sampling [27] Therefore, we refer to 3 phases of decay, the first being months 0-2 (which for HIV DNA we do not estimate), the second 2-12 months (which for HIV DNA we approximate with the 0-1 year phase), and the third from 12 months to 8 years.

Given these definitions, we found intact HIV DNA decayed faster than defective HIV DNA in children with viral load suppression during the first year on ART (phase 2 p=0.0003, **Fig 2B**), and during 1-8 years on ART (phase 3 p=0.02, **Fig 2B**). This demonstrates differential decay of intact and defective proviruses as has been observed in adults and children in many different cohorts with other timelines and HIV subtypes.

As a sensitivity analysis on our suppression criterion choice, we estimated intact HIV DNA decay rates with different criteria for suppression (**S8 Fig**). There were 7 children who maintained HIV RNA < 300 copy/mL for the entire study and also had observed intact DNA data, though only 1 of these children had 3 longitudinal time points, the rest having a single time point. Intact half-life estimated in these children was 22 months, with 95% confidence intervals ranging from 8-infinite. There were 17 children who maintained HIV RNA < 1000 copy/mL for the entire study and also had observed intact DNA data, with 9 of these children having 2 or more longitudinal time points. Intact half-life estimated in these children was 55 months, with 95% confidence intervals ranging from 16-infinite. Overall, this analysis showed mean estimates in more stringently defined suppression groups did not disagree with the initial criteria (**Fig 2**), but estimates were less imprecise due to low numbers.

A strength of pNLME modeling is that a population average is rigorously defined while individual rates are estimated for each participant [49]. Thus, we can check whether the population trends apply to individuals. Indeed, the same kinetic patterns revealed across the population (**Fig 2C**) were also seen for certain individual participants with extensive longitudinal sampling (**S6 Fig**). We did not observe any differences between HIV provirus kinetic rates stratified by participant sex, age at ART initiation, or ART regimen. However, related to the results above, on the individual level we also saw that faster decay of intact HIV proviruses was significantly correlated with faster decay of defective HIV proviruses in the first year on ART (rho=0.5, p=5e-4) (**S7 Fig**).

## Relationships between immune markers and HIV reservoir kinetics

We sought to determine whether several important plasma immune markers related to cell proliferation and/or killing could be related to HIV reservoirs dynamics. Thus, we measured longitudinal levels of interleukins IL-2, IL-7, IL-15, and Granzyme B (GzB) in N=30 children who had at least 3 measurements of HIV proviruses after 1 year on ART. Our hypotheses were that elevated levels of the proliferative cytokines IL-7 and IL-15 would associate with slower decay of defective HIV proviruses and/or expansion of CD4 count, and that elevated levels of IL-2 and GzB which are biomarkers of immune activation, would associate with faster decline of viral load and/or intact vs. defective HIV proviruses because increased immune recognition may preferentially deplete cells harboring intact but not defective provirus. Immune marker kinetics in our cohort followed patterns of early decay followed by a stable setpoint (GzB), or a stable level throughout ART (IL-2, IL-7, and IL-15) (**Fig 3A**). Within participants, early (at ART initiation) and late (average >1 year of ART) levels of IL-2 and IL-15 were strongly correlated (rho>0.7) and IL-7 was moderately correlated with these two (rho>0.4) (**S9A Fig**).

There were correlations between immune markers and viral kinetics: higher baseline GzB was associated with faster phase 1 RNA decay Spearman rho=0.6, p=0.01, **Fig 3B**). Higher baseline IL-15 levels were associated with faster phase 2 intact HIV provirus decay (Spearman

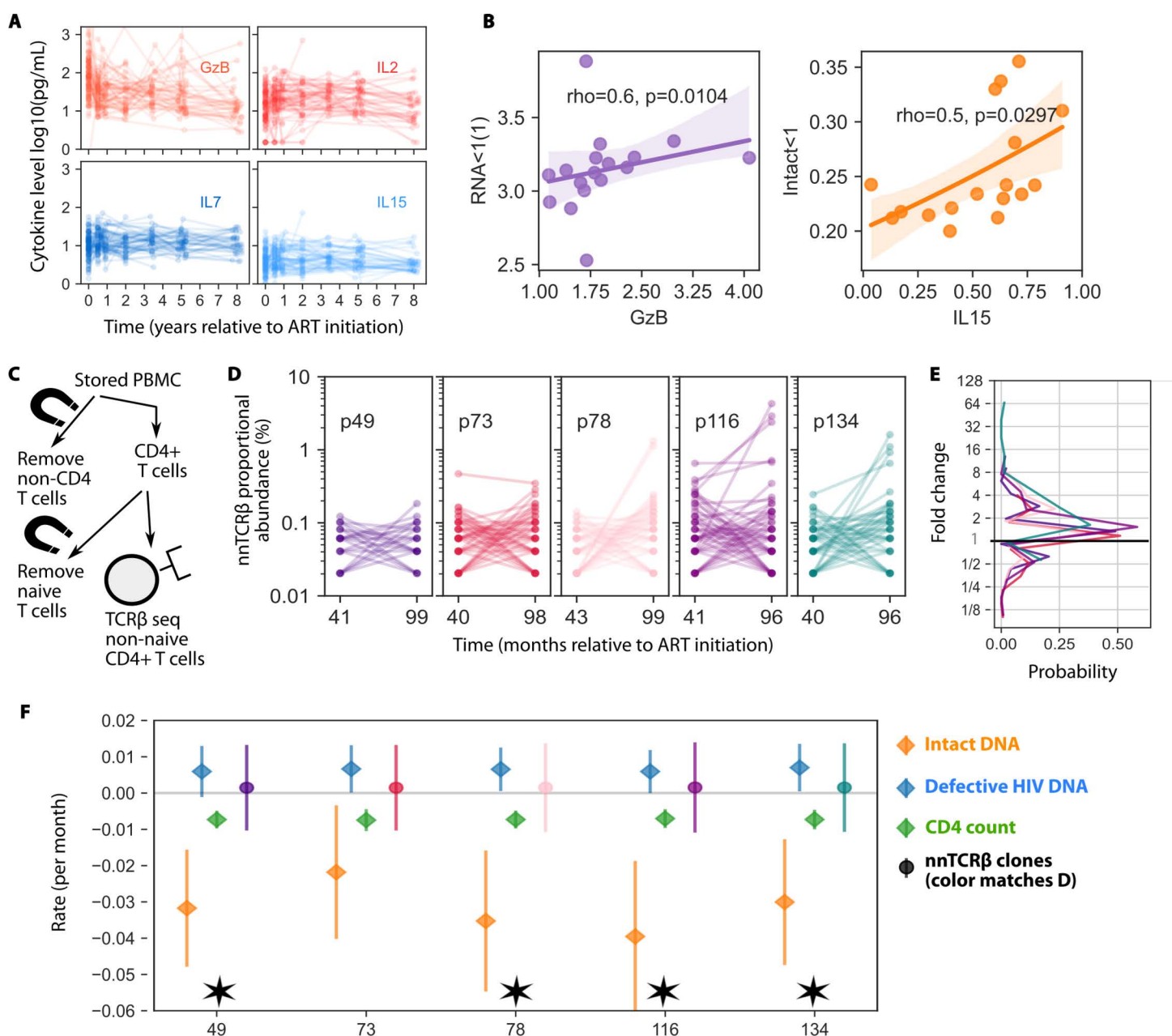

**Fig 3. Comparison of HIV kinetics to immune marker dynamics and CD4+ T cell clone dynamics.** *A) Kinetics of* immune markers *measured in 30 OPH participants. B) Baseline GzB and IL15 (at ART initiation) associated with very early viral load (<2 months) and early (<1 year) intact decay, respectively. Line/shading is linear regression and 95% CI. Spearman correlation coefficients and p-values are noted but are not significant after multiple comparisons correction (significance would require p<0.003). All comparisons shown in* S7 Fig. *C) Schematic of TCR sequencing after magnetic bead isolation of non-naive CD4+ T cells. D) Paired longitudinal nnTCRβ clone kinetics from five participants from clones observed at both timepoints (roughly 40 to 96 months after ART) after downsampling to lowest sample size across participants (Methods). E) Histogram of fold-changes from paired persistent clones. Colored lines match individuals in D. F) Mean (dot) and 95% CI (line) rate for nnTCRβ kinetic change rates compared against individual HIV DNA and CD4 count rates from the matching participants. Asterisks for 4 of 5 participant IDs denote intact HIV DNA rate is significantly faster than other rates as determined by no overlap in 95% CI. Intact proviruses tend to decline whereas the total is stable and expansion and contraction of typical TCR clones is balanced.*

rho=0.5, p=0.036, **Fig 3B**). We highlight these associations because of their relative strength (rho>0.5) but trends do not achieve p<0.001, a conservative Bonferroni significance threshold given the number of comparisons (S9B Fig).

## Relationship between CD4 clones and HIV DNA kinetics

We next sought to characterize CD4+ T cell clonality and compare with HIV provirus kinetics, with the hypothesis that carrier cell homeostasis predominantly drives the dynamics of HIV proviruses [33]. From PBMC samples from 5 participants with >5 longitudinal HIV provirus measurements (**S5 Fig**), we selected the 40 and 96 month time points (**Fig 1C**), isolated non-naïve CD4+ T cells, and obtained T cell receptor β chain repertoires (nnTCRβ) to estimate clonality from the cell types that most commonly harbor HIV proviruses [31,50] (**Methods** and **Fig 3C**).

Sample sizes (number of cells sequenced) ranged from 4,904 to 61,288 sequences. The fraction of clones that were unique ranged from 70-90%, with richness (number of unique clonotypes) increasing with the number of cells sequenced. However, in larger samples, the same clones were more likely to be repeatedly found (**S10 Fig**) such that richness showed some signs of saturation. For fair comparisons, we downsampled all data sets to the smallest sample size of 4900 cells. From these resampled data, we observed roughly 1-3% of clonotypes persisting through the two time points. Persistent clonotypes were typically small (1/1000 of cells or fewer), though some reached proportional abundances of >1/100 cells in the sample.

These clonotypes expanded and contracted (**Fig 3D**), with a distribution of fold changes that roughly symmetric and centered just above 1. This means that contractions and expansions are generally balanced. It was slightly more likely to observe fold changes greater than 8 vs less than 1/8 (**Fig 3E**), which may mean that large expansions are more common than large contractions, but sampling could drive this finding as contractions are effectively bounded by incomplete sampling.

We also computed the rate change in abundance for each paired clone (**Methods**) and compared nnTCRβ kinetic rates to individual decay rate estimates for HIV provirus and CD4 count from these five participants. There was considerable overlap between estimated decay rates for CD4 count, defective HIV DNA, and nnTCRβ clones, but mean intact HIV DNA decayed faster in all individuals, with significance (no overlap of confidence intervals) found in 4 of the 5 individuals (**Fig 3F**). These results imply that defective DNA kinetics resemble CD4 count and typical non-naïve CD4+ T cell kinetics whereas intact DNA is cleared more rapidly. These findings in **Fig 3F** indicate that there is a balance where some CD4+ T cell clonotypes expand and contract such that the total level tends to stay stable (as CD4 count). The total level of defective proviruses also remains stable, but intact proviruses tend to decline.

## Modeling HIV DNA during RNA viremia

To determine the influence of HIV viremia on HIV DNA, we extended our prior HIV reservoir models [28,51–53] to describe proviral decay while including infected cell creation due to viremia (**Fig 4A**). The model includes intact and defective proviruses and, based on our results here (and prior modeling/analysis of multiphasic decay [54–56], the model includes three phases which are modeled by infected cell compartments $I_1, I_2, I_3$, each with their own clearance rate $\theta$. These clearance rates importantly encompass the net balance of proliferation and death and viral reactivation [51]. First phase infected cells $I_1$ are created proportionally to viral load with a rate $b$. A small fraction of these cells can transition to phase 2 and 3 states with rates $\phi_1$ and $\phi_2$ (all equations in **Methods**). Infected cell compartments and rates are superscripted ($i$, $d$) for intact vs defective with rates allowed to be different.

The model is fit again with pNLME in Monolix but now viral load is treated as a "regressor", an external force that can drive the infected cell dynamics (**Methods**). Therefore, we could fit the model to a total of 82 participants, the 55 participants who were suppressed and had CS-IPDA data as well as 27 participants who had substantial viremia but also CS-IPDA

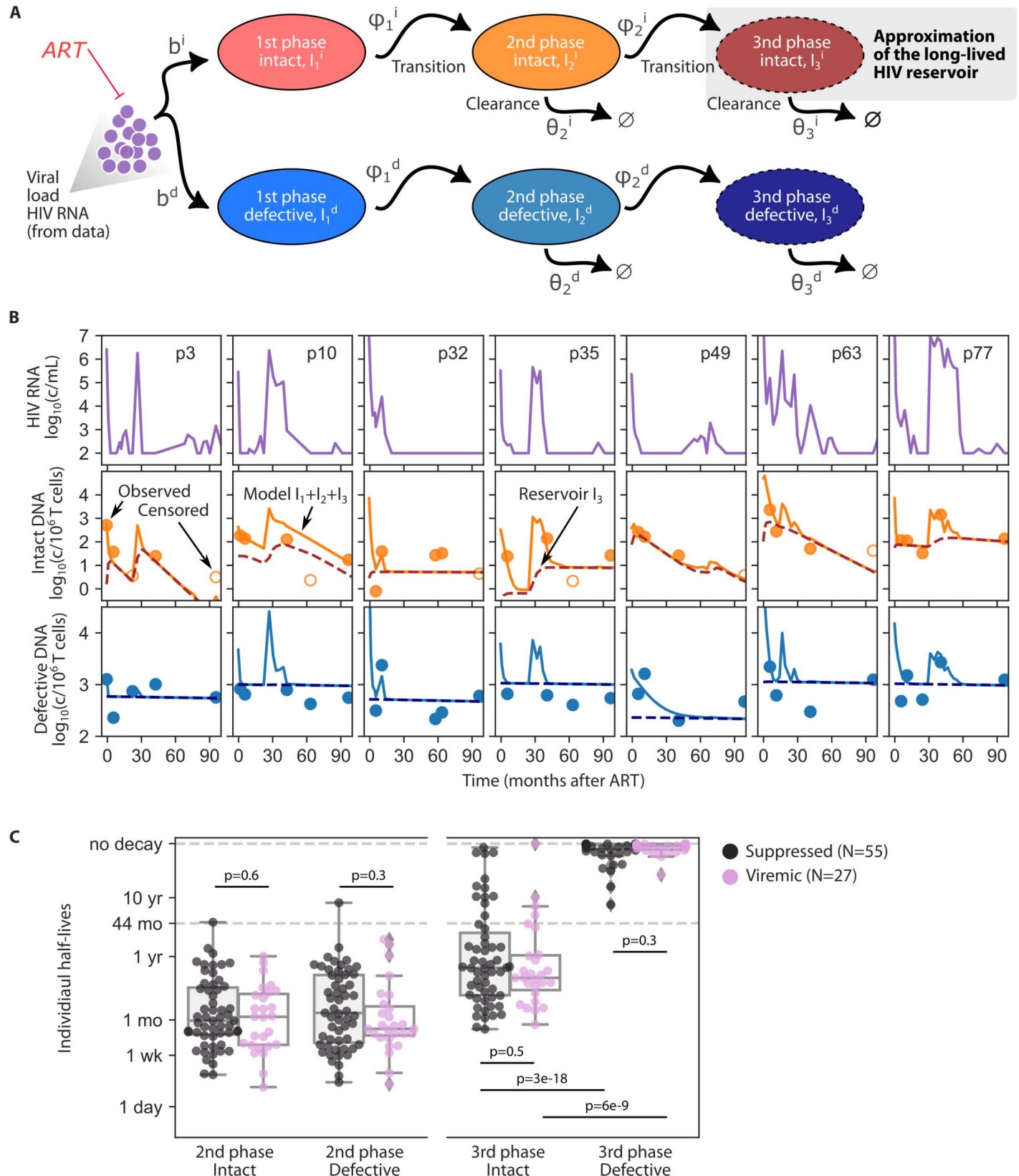

**Fig 4. Modeled kinetics including viremia allow comparison with viremic and suppressed subset.** *A) Model schematic to link HIV RNA viremia and creation of HIV DNA (Phase 2 and Phase 3 or shorter and longer-lived, Eq 1). B) Seven example model fits (all in S1 Data) where each column shows a participant, and the rows*

*show HIV RNA and observed intact and defective HIV DNA (dots) vs model output of total HIV DNA ($I_1 + I_2 + I_3$, solid line) and Phase 3 long-lived HIV DNA ($I_3$, dashed line). C) Box plots of individual half-life estimates for intact and defective HIV DNA for Phase 2 and Phase 3 split by suppressed (N=55) and viremic (N=27) participants. P-values indicate one-sided Mann-Whitney U-Test.*

measurements. Observed levels of intact and defective HIV DNA were compared to the total of the model as $I_{tot} = I_1 + I_2 + I_3$. By adjusting model parameters, we found model output could approximate observed intact and defective provirus levels (7 examples in **Fig 4B**; all participants with 3 or more longitudinal HIV DNA observations are shown in S1 Data). Residual squared errors were generally below 50% indicating parameters are properly estimated and residual sum of squares (RSS) error normalized to number of data points was generally below 0.1 log indicating a relatively successful model fit, albeit with low numbers of observed time points for some participants.

### Modeling suggests DNA creation during viremia is mostly transient

A key model population for HIV cure is $I_3^i$, the long-lived infected cells harboring intact HIV DNA, which we envision as an approximation of the rebound-competent latent reservoir (**Fig 4A**). We assume that during viremia there is creation of infected cells that will eventually enter this reservoir, but we make no assumptions about the exact biology governing this mechanism. However, our model estimates that the reservoir creation rate must be very low. In **Fig 4B** we show the total infected cells (solid lines) as well as the Phase 3 long-lived infected cells (dashed lines). The total matches the observed data (circles), including transient rises concurrent with viremia, but the long-lived cells are not strongly increased. Most DNA creation during these periods is in shorter lived cells that then are quickly cleared once ART resuppression is achieved.

### Similar decay rates for both viremic and suppressed populations

Notably, the best fit model estimates for all clearance half-lives ($\theta$) were not significantly different between the suppressed and viremic participants for any infected cell type (Phase 2 and 3 half-lives for intact and defective HIV DNA, Mann-Whitney p>0.3 for all comparisons, **Fig 4C**). As in the prior analysis, phase 3 half-lives were different between intact (population mean 31 months) and defective (population mean 990 months) HIV DNA (p<1e-8, **Fig 4C**). These results suggest that differential decay of intact and defective proviruses also applies to the participants who had some viremia and that long-term HIV DNA half-lives are not slower in viremic vs suppressed participants.

## Discussion

Here we analyzed an early treated pediatric cohort of children living with HIV (CWH) from Nairobi, Kenya. Viremia was observed in many children, emphasizing that durable suppression is difficult and reinforcing the crucial need for cure in this population [57]. Among the suppressed subset of CWH, we estimated the longitudinal kinetics of HIV viral load (RNA), CD4 count, and intact and defective HIV DNA, which were quantified using our cross subtype intact proviral DNA assay (CS-IPDA) and population nonlinear mixed effects (pNLME) modeling. HIV viral load decayed rapidly in two phases until undetectable by year 1. CD4 counts expanded early and then contracted, similarly to children without HIV. Intact DNA decayed more rapidly than defective DNA in early (<1 year) and long-term (1-8 year) treatment windows, with similar patterns and half-lives to those previously estimated from pediatric [6,58–63] and adult [48,64,65] cohorts. We determined that 4 relevant immune markers for

proliferation and/or killing were not obviously correlated with decay rates, and that non-naïve CD4+ T cell clone kinetics resembled those of defective but not intact HIV DNA, suggesting intact proviruses could influence the death rate of their carrier cells or less frequently become integrated into very long-lived cells than other proviruses. Finally, we developed a model to study the viremic subset of CWH, finding that HIV RNA can be linked to transient rises in HIV DNA, but that viremia (due likely to treatment interruptions) do not strongly influence long-term reservoir size and kinetics.

Although HIV RNA and DNA kinetics in CWH starting ART are well documented [6,27,58,66–69], only recently have pediatric reservoir studies begun to distinguish between intact vs defective proviruses [19,27,62]. Importantly, cross-subtype multiprobe intact proviral DNA assays such as ours and others are needed to understand reservoirs in African children who often acquire subtype A, D, C or circulating recombinant (AE) viruses [19,70,71]. Our results solidify that differential decay of intact and defective proviruses is present in children with non-subtype B infection over very long time scales [72], and that decay rates overlap prior estimates in adults [22,24,25,64]: mean half-lives were 3 months for intact vs 9 months for defective in the first year of ART and then 22 months for intact vs no decay for defective (101 month doubling time) after 1 year on ART.

We also observed that CD4 count (CD4+ T cells per μL) expanded during the first year of ART and then declined afterwards, roughly similar to cell dynamics observed in historic cohorts of children without HIV. An intriguing but likely minor consequence of declining CD4 counts is that given we measure HIV DNA per million T cells, the total body burden of HIV DNA might be declining even faster than the rate we measure per million cells.

To gain mechanistic insight into immunological factors that could contribute to HIV DNA decay, we also examined immune marker dynamics from 30 participants [34,35]. Rather than a broad exploratory analysis, we studied four markers that have been related to key factors for reservoir persistence including T cell proliferation, latency reversal, and immune activation/killing (IL-2, IL-7, IL-15, and Granzyme B) [73]. We did note a trend associating higher baseline IL-15 and faster early intact decay. In humans, the IL-15 superagonist N-803 appears to raise NK cell levels [73]. In early treated CWH, reservoir seeding was found to relate to NK cell responses [74]. Thus, although we hypothesized that IL-15 might associate with reservoir expansion through latently infected cell proliferation, it may be that NK cell mediated killing counteracts or overcomes those expansions. However, this trend was not statistically significant after correcting for multiple comparisons, so future studies with larger cohorts will be needed.

Non-naïve T cell receptor (nnTCRβ) sequencing in 5 children allowed us to interrogate the "passenger" hypothesis: that the natural physiology of CD4+ T cells, which involves both cell death and proliferation, is primarily responsible for the dynamics of HIV proviruses [33]. We sorted non-naïve CD4+ T cells, the putative carriers of most HIV proviruses. Between 40 and 96 months after ART, clonal expansions and contractions were roughly symmetric. There was some indication that large expansions were more common than large contractions, but interpretations are difficult given that contraction size estimates are naturally underestimated when clones become undetectable. Still, averaging across these size change distributions, nnTCRβ clonotypes appear very stable, with the most common events being small clonal expansions balanced by small clonal contractions. These clonal dynamics agree with observations that individual HIV clones both wax and wane (in adults [32]) and persist for up to 9 years (in CWH [75]). Combining all clonotypes, our average estimated nnTCRβ clone kinetic rates overlapped those of bulk CD4 count and defective HIV kinetics estimated from the same participants. However, intact HIV provirus dynamics do not appear to be easily explained by passenger kinetics alone, implying additional forces like negative selection on intact proviruses are present in children, as is observed in adults [33,76,77].

One of the major takeaways from our study is the consequences of treatment interruptions in pediatric HIV cohorts. Consensus in the field indicates well-monitored analytical treatment interruption (ATI) studies are largely safe for participants and resuppression appears to return HIV DNA levels to pre-ATI levels [42,78–82], though transmission risk must be considered and some changes to reservoir diversity may be expected [82]. In detailed macaque experiments, barcoded viruses did re-seed reservoirs during ATI [83]. Our cohort consisted of some children who were randomized to ATI at year 2, and others who experienced unplanned treatment interruptions [8]. To assess the impact of treatment interruptions on reservoirs, we modeled HIV DNA inclusive of periods with substantial HIV RNA viremia. We recapitulated observed HIV DNA with a model that described shorter and longer lived (Phases 1, 2, and 3) infected cells. The model projects that the majority of HIV DNA generated during periods of non-suppressed viremia is in shorter lived cells. This implies that HIV DNA transiently rises during periods of viremia but largely returns to baseline within months after re-suppression. Therefore, our model projects a mechanistic basis for why treatment interruptions are not significantly influential on longer term reservoir size and dynamics.

Relatedly, we showed that stringent inclusion criterion of undetectable viremia (as typically used for reservoir studies) might be unnecessarily strict given the heterogeneity of participant data, assay variability and/or noise underlying reservoir data. Indeed, values of intact and defective HIV DNA were not significantly different at timepoints with viral loads <3000 copies/mL vs completely undetectable (<150 copies/mL). Moreover, intact and defective HIV DNA half-lives were similar between viremic and suppressed individuals, shown both via sensitivity analysis (RNA suppression criteria of <300, <1000, and <3000 copies/mL in original decay estimates) as well as using our novel modeling inclusive of viremia. In both prospective and retrospective studies of children with HIV, investigators may consider balancing the scientific rigor imposed by stringent inclusion criteria with the practical realities of children with HIV not always maintaining complete viral suppression while taking ART.

There are several recent pieces of evidence informing reservoir creation that may relate to our work here. First, from macaque studies, we know that rebound competent reservoirs are created extremely early but also appear to saturate [84]. HIV DNA set points are also established extremely early in humans [85]. Of many clinical variables tested in human ATI studies, viral load area under the curve during primary infection is the most correlated to the timing of viral rebound after ATI [86,87]. Relatedly, timing of ART initiation after acquisition seems to delicately dictate rebound kinetics after ATI – in a meta-analysis of 124 NHP animals from 10 independent studies, investigators found extremely early therapy actually increased rebound peak viremia until week 3, after which delaying ART led to increases in post-rebound setpoint viral levels [88]. Meanwhile, proviral sequencing suggests a continuum of seeding times, with a bias towards times closer to ART initiation [53,89,90]. Putting this evidence together with ours, we hypothesize that reservoir seeding pre-ART saturates such that qualitatively different processes of reservoir turnover may occur during treatment interruptions [91].

We propose that our new approach to incorporating active viremia into models of HIV DNA decay could help re-analyze prior cohorts. Because viremic CWH would be excluded from typical reservoir studies, reservoir decay properties in important CWH is unknown (e.g., those with known adherence that still struggle to suppress viremia [38]). We were able to directly compare the half-lives of infected cells estimated in suppressed and viremic participants, finding that intact proviruses decay faster than defectives in both groups. Moreover, the half-lives of infected cells that ultimately persist during ART were not significantly different between suppressed and viremic participants, suggesting that treatment interruptions do not strongly influence the long-term kinetics of reservoir persistence, though it is unknown if this holds true for other populations.

There are other limitations to our work. Sparser data meant confidence intervals in estimated decay rates for participants after 1 year of ART were wide and possibly inclusive of growing intact reservoirs at later stages. Because blood volumes from children are small, we measure HIV DNA per million T cells, vs per million CD4+ T cells as in some other studies. Yet our decay rate estimates do agree with most prior values, suggesting this denominator does not strongly bias results. Losing individuals to follow up is a typical problem for long-term studies and can also bias results. Thus, our original decay estimates apply to the types of individuals who can maintain long-term suppression – but our work highlights this caveat for most if not all other HIV reservoir studies. By including children who had viremia and found their long-term HIV DNA decay rates were not different, our work potentially expands the applicability of many studies. The model linking HIV RNA and DNA is based on prior work in adults but fits our pediatric data, both in the suppressed and viremic subsets. However, the assumption that a long-lived compartment is created continuously during viremia may not be accurate, as other studies suggest sequences might be more likely to be archived into the reservoir near the time of ART [53,89,90,92].

In the future, we hope to collect sequencing data to help interpret the results. For instance, it is unclear if treatment interruptions led to any drug resistance evolution and we cannot address whether this factor contributes to heterogeneity in decay patterns across individuals, though the present decay estimates are not notably more heterogeneous than in prior cohorts in children [10] or adults [64].

Studying children is particularly important because the early-life immune system might better-respond to curative interventions. Children have lower overall immune activation [2] and pediatric reservoirs appear less inducible [93], which could theoretically help or hinder therapies attempting to achieve a functional cure. Our findings suggest reservoir creation during treatment interruptions is relatively transient, but that over 8 years intact HIV DNA decays very slowly, and it is unknown what occurs beyond this time frame in CWH. Thus, interventions that can improve this trajectory would be critically important for the long-term prognosis of these children and models like this will be essential for assessing those interventions.

## Materials and Methods

### Ethics statement

The study was approved by the University of Washington Institutional Review Board (IRB) and the Kenyatta National Hospital (KNH)/University of Nairobi (UoN) Ethics and Research Committee for NCT00428116. Oral informed consent was obtained from each adult participant enrolling in the study; written informed consent was then obtained from the subset of those completing pediatric testing. Older children whose parents agreed provided written assent for the HIV test. At the time of this study, children ≥16 years (or emancipated minors > 14) could provide independent consent for HIV testing, while others required parental consent; these older ages were not included in this study.

### Study cohort and sample collection

This study used archived samples collected between 2007-2017 from the Optimizing Pediatric HIV (OPH) cohort in Nairobi, Kenya. Procedures for enrollment and follow-up of this pediatric cohort have been described elsewhere [8]. Briefly, ART-naïve infants <12 months old were enrolled at prevention of mother-to-child transmission of HIV clinics and pediatric wards, initiated ART, 42 were randomized into an ATI study (NCT00428116), and follow up was for ~10 years, including collection of blood samples that were separated into plasma and peripheral blood mononuclear cells (PBMCs), which were frozen or cryopreserved until use.

## HIV RNA and DNA quantification

HIV RNA was previously quantified with the Gen-Probe HIV RNA assay that had a lower limit of detection of 150 copies/mL [8]. Data below those levels were set at 75 copies/mL, the midpoint of 0 and the threshold. Total and intact HIV proviruses were quantified using the cross-subtype intact proviral DNA assay (CS-IPDA) as described previously [19,36]. Briefly, CS-IPDA was performed in triplicate. If no intact HIV proviruses were detected, additional replicates were performed until either intact proviruses were detected or a minimum of 1e5 cells were interrogated. DNA shearing rates are measured by the RPP30 reference assay, and both total and intact DNA levels reported if shearing <40%.

## Population nonlinear mixed effects (pNLME) model fitting

We used population nonlinear mixed effects modeling (pNLME) implemented in Monolix (www.lixoft.eu) to estimate population rates and individual rates for each participant. In this approach each data type ($Y$) for each individual $i$ at time $j$ is modeled with exponential dynamics such that $\ln(Y_{ij}) = \theta_i^Y \times t_{ij} + Y_0 + \in_Y$. The initial condition is $Y_0$ and the noise is modeled by $\in_Y \sim \mathcal{N}(0, \sigma_Y^2)$, where $N$ denotes a normal distribution. This effectively leads to a log-normal distribution of estimated measurement error for the data type with its standard deviation $\sigma_Y$. The only exception to this model was the model for viral load within the first year of therapy which used a biphasic model: $Y_{ij} = \in_Y \left[ Y_1 \exp\left(\theta_i^{V1} \times t_{ij}\right) + Y_2 \exp\left(\theta_i^{V2} \times t_{ij}\right) \right]$. The pNLME approach also assumes that each individual-specific parameter $\Theta_i^Y$ is drawn from a probability distribution with the median population parameters (fixed effects) $\Theta^{Y_{pop}}$ and the random effects that define the individual-specific value: $\eta_i^Y \sim \mathcal{N}(0, \Omega)$, where $\Omega$ is the covariance matrix (admitting each parameter $p$'s variance in the diagonal elements $\Omega_{p,p}$ and the covariance between two different parameters $p1$ and $p2$ as the off diagonal elements $\Omega_{p1,p2}$). Parameters were modeled assuming log-normality for initial conditions, and normality for decay rates. Importantly, decay rate constants $\theta_i^Y$ were allowed to be positive or negative to allow for decay or expansion. Monolix calculates the Maximum Likelihood Estimation (MLE) of the measurement error standard deviation $\sigma_Y$, the fixed effects vector $\Theta^{Y_{pop}}$ and the elements of matrix $\Omega$ using the Stochastic Approximation of the Expectation Maximization (SAEM) algorithm embedded in the Monolix software. Individual parameters were selected using the mode of the conditional distribution $f(\Theta_i \mid Y_{ij}; \Theta_{MLE}^{Y_{pop}}, \Omega_{MLE})$ constructed by the MCMC algorithm in the Monolix software.

## Time points for immune marker analysis

We selected 30 participants from the OPH cohort with at least 3 HIV DNA time points after 1 year of ART. We measured immune markers (Meso Scale Discovery custom 10-multiplex panel) from plasma samples at ART initiation (baseline), 6 months after ART initiation, and at HIV DNA time points (**Fig 1**). For 6 missing baseline measurements, we selected a sample within -3 to +3 months of starting ART. For 20 missing HIV DNA time points, we selected a plasma sample within 6 months of that time point. For >40 month missing time points, we selected the closest time point before or afterwards, for <24 month missing time points, we selected the closest time point afterwards.

## Non naïve CD4+ T cell TCRβ sequencing

We identified 14 CWH from the suppressed subset with at least 3 HIV DNA time points after 1 year of ART. From this group, we selected 5 donors due to multiple available quality aliquots, participant IDs 49, 73, 78, 116, and 134. We isolated CD4+ T cells from total PBMCs using a negative selection magnetic bead kit (Miltenyi Biotec #130-096-533) then

used a negative selection naïve T cell magnetic isolation kit (Miltenyi Biotec #130-094-131) to separate non-naïve cells. The non-naïve cells are therefore PBMCs that are negative for: CD8a, CD14, CD15, CD16, CD19, CD36, CD56, CD123, TcRγ/δ, and CD235a AND positive for at least one of the following: CD25, CD45RO, and HLA-DR. Thus, it is possible they are CD45RO- but if so then they have either CD25 or HLA-DR as an activation marker. From these cells, DNA was isolated (Qiagen #69504) and TCRβ sequencing (to ultra-deep scale) was performed by Adaptive Biotechnology.

## TCRβ clone dynamic calculations

To assess the rate of change in TCRβ clone size over time, we analyzed paired samples from 5 CWH taken at approximately 40 and 96 months after ART initiation. The proportional abundance of each TCRβ clone $i$ was calculated as it's observed abundance divided by the total number of cells $p_i = a_i / n$. To standardize comparisons across samples, resampled abundances were defined using multinomial sampling to a common sample size, $\hat{a}_i = M(p_i, s)$ where $s$ =4900 was the lowest sample size across all 10 time points. Then, resampled abundance data from paired time points were merged on TCRβ amino acid sequences. The change in clone size over time was quantified by calculating the rate of change ( $k_i$ ), defined assuming exponential kinetics divided by the time interval $\Delta t$ between the two samples in months:

$$\ln\left(\frac{\hat{a}_i\left(t_2\right)}{\hat{a}_i\left(t_1\right)}\right) / \Delta t = k_i .$$

## Mathematical model including HIV RNA levels into HIV DNA dynamics

We developed a model that grows from a basic viral dynamics formulation to describe HIV RNA and DNA simultaneously during periods of viremia and ART. Previously, models were successfully fit to HIV RNA decay data in PWH who initiated ART by assuming the existence of 3 types of infected cells with different lifespans [28,55,56,94]. Such a model can be expressed in several mathematical formulations, with slightly different mechanistic interpretations, but here we use:

$$\begin{aligned}
\dot{S} &= \alpha_s - \delta_s S - \beta S V \\
\dot{I}_1 &= \beta S V - \theta_1 I_1 - \phi_1 I_1 + \xi_2 I_2 + \xi_3 I_3 \\
\dot{I}_2 &= \phi_1 I_1 - (\theta_2 + \phi_2) I_2 \\
\dot{I}_3 &= \phi_2 I_2 - \theta_3 I_3 \\
\dot{V} &= \pi I_1 - \gamma V - \beta S V
\end{aligned}$$

(1)

where overdot denotes time derivative. Here, $\alpha_s$ is the source of susceptible cells, $\delta_s$ the natural death rate of these cells, $\theta_s$ the clearance rate of each infected cell type $s$ (which is the net rate from a balance of forces including proliferation, viral cytolytic death, and immune system killing and potentially others), $\phi_s$ represents a transition rate into a longer-lived infected cell type, $\xi_s$ is the rate of reactivation into the $I_1$ compartment, $\pi$ the rate of viral production and $\gamma$ the rate of natural viral clearance. Roughly, we imagine $I_1$ as short-lived actively infected cells producing virus (approximately 1 day lifespan), $I_2$ as medium-lived infected cells (with approximately 6 week lifespans and potentially in a pre-integration state as derived via prior modeling [55]), and $I_3$ as long-lived/proliferating infected cells, approximating latency and resulting in very slow clearance (multi-year half-life).

Now, we make several simplifications based on the sampling frequency of the data and the scientific question we are addressing: what is the contribution of transient viremia to the HIV reservoir via treatment interruptions? The first simplification is that reduction of virions due to infection is negligible (i.e., the $\beta S V$ term in the $\dot{V}$ equation can be ignored). The second is that viral dynamics are fast enough compared to the rest of the system such that we can use a quasi-steady state approximation $I_1 = \gamma V / \pi = b\mathcal{V}[t]$. Finally, the replicative dynamics relating susceptible cells, $I_1$ cells, and virus can be ignored in favor of directly utilizing the observed HIV RNA data $\mathcal{V}[t]$. This results in a highly simplified model

$$\dot{I}_2 = \phi_1 I_1 - (\theta_2 + \phi_2) I_2$$
$$\dot{I}_3 = \phi_2 I_2 - \theta_3 I_3$$

(2)

We separately model intact and defective HIV DNA, so we have a copy of this system for intact and defective (noted with superscripts in **Fig 4A**).

Because of our main simplification, we do not directly model target cell limitation, nor do we consider reactivation into $I_1$. However, the emergent viral loads we impute into the model likely are implicitly affected by these phenomena. Our approximation could be stated that the dynamics of $I_2$ and $I_3$ cells are an epiphenomenon whose dynamics can be effectively ignored in modeling the main viral dynamics. As evidence, many viral dynamics models fit to primary infection data without these compartments.

It is unclear whether infected cells with longer lifespans are created directly by infection or transition from a shorter lived compartment (our choice here). Additional data would be required to disambiguate these approaches. But, we did assess a mathematically different but ultimately very similar model in which the $I_2$ and $I_3$ compartments were generated independently in parallel via a fraction $\tau$, leaving the equations as $\dot{I}_2 = \tau b\mathcal{V}(t) - \theta_2 I_2$ and $\dot{I}_3 = (1-\tau) b\mathcal{V}(t) - \theta_3 I_3$. This led to generally similar results because the $b$ and $\phi$ terms are more rapid than clearance rates. Additionally, we explored a nonlinear reactivation term such as $\xi_3 I_3 \mathcal{V}[t]$ such that increasing viremia increases activation into the more active pool [95]. However, this model became numerically unstable and often led to negative numbers due to the imputed viral load changes, so we opted to not use it in our more simplified implementation.

### Fitting the model inclusive of viremia

To fit the model, we again used population nonlinear mixed effects modeling in Monolix where viremia was input as a "regressor" or external variable that changes the infected cell populations. We estimated rate parameters for each type (intact and defective) $b, \theta_2, \phi_1, \theta_3, \phi_2$ and the initial condition $I_2(0)$. The second initial condition was derived from a quasi-static approximation: $I_3 = \phi_2 I_2(0)/\theta_3$. Parameters were assumed to follow log-normal distributions and noise was assumed to be constant on the log-scale. Intact HIV DNA that was not detected was set to censored value at 1/total cells assayed.

### Statistical analysis

Statistical tests are noted in each use. Z-test is the difference of means ($\mu$) divided by the square root of the sum of each standard error squared, i.e., $Z = (\mu_X - \mu_Y) / (s_X^2 + s_Y^2)^{1/2}$, where standard error is the standard deviation divided by the square root of the sample size $s = \sigma / \sqrt{N}$. P-values were computed by the cumulative probability distribution of a standard normal distribution at value Z. Whenever presented, box plots indicate medians, interquartile ranges (IQR), and whiskers indicate 1.5x IQR.

## Supporting information

**S1 Table. All rates for each population parameter estimated from the suppressed OPH subset using an exponential decay model (see Methods).** Rate is the mean population parameter rate (per month), SE is the standard error of each population parameter (standard deviation divided by square root of sample size, $\sigma / \sqrt{n}$), and RSE is the percent residual squared error, the ratio of SE to Value x 100%, which indicates better fit for lower values. Data types are HIV RNA, Intact and Defective HIV DNA, and CD4 count, where <1 and >1 indicate the period before and after 1 year of ART. Negative rates indicate decay whereas positive indicates expansion.
(XLSX)

**S1 Fig. Number of follow up time points for HIV RNA, CD4 count and HIV DNA (intact and defective are the same).**
(TIF)

**S2 Fig. HIV DNA in suppressed vs viremic subsets.** A) Intact and B) defective HIV DNA levels in the suppressed vs viremic subsets (suppressed criteria was HIV RNA levels dropping below 1000 copies/mL within 1 year of starting ART and remaining below 3000 copies/mL thereafter). *n* indicates number of time points in each boxplot. P-values are from one-sided Mann-Whitney U test comparing the boxes with the horizontal line. Box plots show medians, interquartile ranges and 1.5x interquartile ranges.
(TIF)

**S3 Fig. Using data from the suppressed subset, comparison of intact (A) and defective (B) HIV DNA levels when they were detectable (but low, below our defined threshold of 3000 copies/mL) vs undetectable (note assay limit of detection was HIV RNA < 150 copies/mL).** n indicates number of time points in each box. There was no difference between intact and defective HIV DNA levels in participants with low but detectable HIV RNA vs completely undetectable HIV RNA. P-values are Mann-Whitney indicating differences between the two boxes with the horizontal line. Box plots show medians, interquartile ranges and 1.5x interquartile ranges.
(TIF)

**S4 Fig. Relationship between HIV RNA viral load area under the curve (AUC) and final intact HIV DNA level.** We use a trapezoidal numerical integration to estimate the AUC up until the time of the final intact HIV DNA measurement and then compare this to the final measurement. The dot color indicates the final time point, so darker points are later in time and thus have higher AUC. But, there is not a strong statistical relationship, p=0.3 noted in panel title, and note p=0.9 for the identical analysis for defective HIV DNA.
(TIF)

**S5 Fig. Longitudinal levels relative to birth of bulk CD4+ T cell counts suppressed subset OPH participants vs 3 historic pediatric studies that measured CD4+ T cell counts in children not living with HIV.** The studies are denoted by the marker, means are represented by dots and vertical lines are 95% confidence intervals as reported in the original studies. The thicker green line indicates the mean value averaged in 6 month windows for the OPH participants, and the thin green lines represent 95% confidence intervals on these estimates. In both OPH participants and historic controls there appears to be a slight rise in the first year of life and a gradual decay afterwards.
(TIF)

**S6 Fig. Individual model fits (deriving from models shown in Fig 2) for 5 suppressed subset OPH participants with at least 5 longitudinal HIV DNA time points.** Top row axes indicate HIV RNA (left) and CD4 count (right). Jagged lines with small dots are observed

data, solid lines are best model fits. Diamonds for intact HIV DNA indicate levels below limit of detection. The population trends hold for individuals.
(TIF)

**S7 Fig. Association between individual kinetic rates for each data type (HIV RNA, CD4 count, intact and defective HIV DNA) and interval (<1 and >1 year of ART, HIV RNA is additionally split into 0-2 and 2-12 month phases noted by (1) and (2)). Data underlying boxes with significant correlation coefficients are shown separately in scatter plots with linear trendlines.**
(TIF)

**S8 Fig. Intact decay rates with different criteria for suppression.** A) Modeling data from participants who had HIV RNA levels below 300 copies/mL at observed time points after 1 year on ART. Left) Viral load levels. Right) Intact HIV DNA levels (colored dots/lines for each participant, matching VL) and mixed effects model decay estimate (black line = population mean, shaded gray = confidence intervals, the values are noted in brackets as [mean, lower 95% CI, upper 95%CI]. There were 7 children who had any intact time points given this criteria and only 2 child with more than 1 longitudinal time point. B) Same plots but HIV RNA < 1000 copies/mL. There were 17 children who had any intact time points given this criteria and 9 children with more than 1 longitudinal time point.
(TIF)

**S9 Fig. Correlations among IL/GzB dynamics and against estimated kinetics rates.** A) Spearman correlation between 4 immune marker levels at baseline (at ART initiation) and during setpoint (average >1yr of ART). P values are noted for each correlation coefficient. B) Spearman correlation coefficient between immune marker levels and HIV RNA, DNA and CD4+ T cell kinetic rates estimated via pNLME models for 18 participants who had rate estimates for all data types. Baseline immune marker levels are compared against rates estimated from early (<1 yr) of ART and setpoint levels are compared against rates from long-term (>1 yr) ART. P values are noted for each correlation coefficient.
(TIF)

**S10 Fig. Relationship between initial sample size (total number of cells sequenced) and richness (number of unique clonotypes) observed in non-naïve CD4+ TCRβ sequencing. Circles indicate 40 month time points and squares indicate 96 month time points from 5 PWH (colors match** Fig 3**).**
(TIF)

**S1 Data. All viremia model fits (7 examples shown in Fig 4) from OPH participants with 3 or more longitudinal CS-IPDA measurements.**
(PDF)

## Acknowledgments

We gratefully recognize the study participants, as well as the clinical and administrative teams from the Optimizing Pediatric HIV (OPH) Cohort.

## Author contributions

**Conceptualization:** Daniel B Reeves, Annukka A R Antar, Lillian B Cohn, Grace John-Stewart, Dara A Lehman.

**Data curation:** Daniel B Reeves, Morgan Litchford, Carolyn S Fish, Anna Farrell-Sherman, Makayla Poindexter, Jillian K Neary.

**Formal analysis:** Daniel B Reeves, Makayla Poindexter, Nashwa Ahmed.

**Funding acquisition:** Daniel B Reeves, Dalton Wamalwa, Lillian B Cohn, Grace John-Stewart, Dara A Lehman.

**Investigation:** Daniel B Reeves, Morgan Litchford, Anna Farrell-Sherman, Makayla Poindexter, Noah AJ Cassidy, Dara A Lehman.

**Methodology:** Daniel B Reeves, Morgan Litchford, Carolyn S Fish, Anna Farrell-Sherman, Noah AJ Cassidy, Jillian K Neary, Joshua T Schiffer, Dara A Lehman.

**Project administration:** Daniel B Reeves, Sarah Benki-Nugent, Grace John-Stewart, Dara A Lehman.

**Resources:** Morgan Litchford, Carolyn S Fish, Anna Farrell-Sherman, Dalton Wamalwa, Agnes Langat, Daisy Chebet, Helen Moraa, Sarah Benki-Nugent, Dara A Lehman.

**Software:** Daniel B Reeves.

**Supervision:** Jennifer Slyker, Lillian B Cohn, Joshua T Schiffer, Julie Overbaugh, Grace John-Stewart, Dara A Lehman.

**Validation:** Daniel B Reeves, Dara A Lehman.

**Visualization:** Daniel B Reeves.

**Writing – original draft:** Daniel B Reeves, Dara A Lehman.

**Writing – review & editing:** Daniel B Reeves, Morgan Litchford, Anna Farrell-Sherman, Jillian K Neary, Annukka A R Antar, Jennifer Slyker, Lillian B Cohn, Joshua T Schiffer, Julie Overbaugh, Grace John-Stewart, Dara A Lehman.

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
