## [Decision Letter · Decision Letter 0]

20 Dec 2024

PPATHOGENS-D-24-02557

Intact HIV DNA decays in children with and without complete viral load suppression

PLOS Pathogens

Dear Dr. Reeves,

Thank you for submitting your manuscript to PLOS Pathogens. After careful consideration, we feel that it has merit but does not fully meet PLOS Pathogens's publication criteria as it currently stands. Therefore, we invite you to submit a minor revision of the manuscript that addresses the points raised during the review process, particularly providing more information on validation of the CS-IPDA, a more clear presentation of the modeling, and additional emphasis on the biological and clinincal relevance of the findings.

Please submit your revised manuscript within 30 days Feb 18 2025 11:59PM. If you will need more time than this to complete your revisions, please reply to this message or contact the journal office at plospathogens@plos.org. Please include the following items when submitting your revised manuscript:

We look forward to receiving your revised manuscript.

Kind regards,

Mary F Kearney

Academic Editor

PLOS Pathogens

Susan Ross

Section Editor

PLOS Pathogens

Sumita Bhaduri-McIntosh

Editor-in-Chief

PLOS Pathogens

orcid.org/0000-0003-2946-9497

Michael Malim

Editor-in-Chief

PLOS Pathogens

orcid.org/0000-0002-7699-2064

**Journal Requirements:**

At this stage, the following Authors/Authors require contributions: Daniel B Reeves, Morgan Litchford, Carolyn Fish, Anna Farrell-Sherman, Nashwa Ahmed, Makayla Poindexter, Noah Cassidy, Jillian Neary, Dalton Walmawa, Agnes Langat, Daisy Chebet, Helen Moraa, Annukka Antar, Jenn Slyker, Sarah Benki-Nugent, Lillian Cohn, Joshua Schiffer, Julie Overbaugh, Grace John-Stewart, and Dara Lehman. Please ensure that the full contributions of each author are acknowledged in the "Add/Edit/Remove Authors" section of our submission form.

https://journals.plos.org/plospathogens/s/submission-guidelines#loc-parts-of-a-submission

Potential Copyright Issues:

- Figure 1A; Please confirm whether you drew the images / clip-art within the figure panels by hand. If you did not draw the images, please provide a link to the source of the images or icons and their license / terms of use; or written permission from the copyright holder to publish the images or icons under our CC BY 4.0 license. Alternatively, you may replace the images with open source alternatives. See these open source resources you may use to replace images / clip-art:

6) Please ensure that the funders and grant numbers match between the Financial Disclosure field and the Funding Information tab in your submission form. Note that the funders must be provided in the same order in both places as well. State what role the funders took in the study. If the funders had no role in your study, please state: "The funders had no role in study design, data collection and analysis, decision to publish, or preparation of the manuscript.".

**Reviewers' Comments:**

Reviewer's Responses to Questions

**Part I - Summary**

Reviewer #1: In this interesting study, Reeves et al. measured plasma HIV RNA levels, CD4+ T cell counts, and the levels of intact and defective HIV DNA proviruses using the cross-subtype intact proviral DNA assay in samples from 120 Kenyan children living with HIV (CWH). The study found that intact proviruses decayed faster than defective proviruses during both the first year of ART and over a longer period (years 1-8). Notably, in children who did not maintain viral suppression, modeling of HIV RNA and DNA revealed that although HIV DNA levels could transiently increase, intact proviral genomes were not significantly affected by brief treatment interruptions.

This study provides valuable insights into the dynamics of intact and defective proviruses during the early years of life in children with HIV. It is particularly strong in its inclusion of data from children who did not remain virally suppressed, a group that represents a significant fraction of CWH. One of the most notable findings is the faster decay of intact HIV DNA compared to defective DNA in both the early (<1 year) and long-term (1-8 years) treatment periods. The authors also suggest that transient increases in HIV RNA, likely due to treatment interruptions, do not have a lasting impact on the long-term dynamics of the HIV reservoir.

Overall, this study is well-conducted, and its findings are both novel and insightful. However, I have several suggestions that may enhance the clarity and impact of the manuscript:

Reviewer #2: In this large study of 120 Kenyan children, the researchers examined the decay of intact proviruses in children who initiated ART in infancy between 1-12 months of age and maintaining different levels of virologic suppression. A cross-subtype intact proviral DNA assay was used to measure intact proviruses. The authors compared the decay of intact and defective proviruses between 55 children with perinatal HIV who had “durable suppression” and 65 who had non-suppressed viremia, supposedly due to ART ART-interruptions, although confirmation of non-suppressed viremia from ART interruption vs. intermittent ART is not verified. With the CS-IPDA, they found differential decay of intact and defectives with half-lives, with intact proviruses showing faster clearance than defective proviruses of 3 and 9 months, respectively. Ongoing decay of intact proviruses was found with a mean half-life of 22 months—while defective DNA showed no decay.

The authors also went on to examine TCRβ clones in a subset of five children—where they discussed that CD4+ TCR β clones wax and waned in patterns more similar to cells with defective proviruses. Importantly, they report on despite ART interruptions and associations with increase in total HIV-1 DNA that intact proviral load did not increase significantly in children-, a finding that is relevant to the field of HIV remission and cure research for children where ART interruptions are required to detect remission.

Overall comments:

This is an important study that fills knowledge gaps on the decay dynamics of cell-associated HIV DNA in perinatal infections, with respect to intact and defective proviruses and as a function of states of virologic suppression. A state-of-the art assay—a cross-subtype Intact proviral HIV DNA assay was used, which the authors’ group has direct expertise. A well-characterized cohort of children are studied. The expertise on the perspectives from adult cohorts is also a study strength.

Critiques:

• The calculated half-lives of intact and defective proviruses and overall study findings need consideration for the limitations of the CS-IPDA, with respect with how the primers were inferred—from historic sequences in Los Alamos—I believe—

o Perhaps the authors can provide any unpublished data on further validation of the CS-IPDA assay from direct analyses of proviral landscape single genome sequences on the cohort—in other words—some validation of the sensitivity and specificity for HIV subtype A , including positive and negative predictive value, as was done for other IPDA assays—Bruner et al., Bucholtz et al and Lee et al.

• In Figure 1—the suppressed subset—while with lower levels of viremia and area under the curve for viremia than the viremic subset, still appear to have ongoing viremia in the 2-3 log range over the time course of treatment, which poses a challenge for more precisely quantifying HIV reservoirs.

• Additionally, within the first two years of ART, and with non-suppressed viremia, episomal HIV DNA can contributes to the total and intact proviral load.

o Please provide an explanation for how the models account for changes in 2LTR circles in the first two years of ART

• CD4+ T cell clonality data was done on a very small subset of the population—N=5, while these data are interesting—it seems insufficient to make meaning conclusions about HIV DNA kinetics

o The conclusion—lines 308-309 seem strong---“We determined that that non--naïve CD4+ T cell clone kinetics could explain defective but not intact HIV DNA, suggesting intact proviruses influence the death rate of their carrier cells”.

• The section on modeling transient rises in HIV DNA during RNA viremia is confusing to this reviewer as the deeply latent (l3) compartment in the model seems not applicable given the persistent viremic state of the participants; it is also unclear what the contribution of selected drug-resistant HIV will perform in this model. Hence the conclusion on page 8, lines 275 and 276 is unclear—“Importantly, model projections indicated that periods of viremia were strongly associated 276 with rises in 2 but not 3 cells (the more deeply latent reservoir)”—how is the deeply latent reservoir inferred here?

Overall. an important set of data to the field of pediatric HIV infection dynamics, but limited by the non-suppressed nature of the cohort, the use of a CS-IPDA with no provision on PPV and NPV on subtype A infection, and reporting on assay performance across 120 participants. The mathematical model on reservoir dynamics is flawed by the mere lack of suppression in the cohort since not all that is intact is likely to seed the reservoir.

Reviewer #3: The study by Reeves et al. Lehman addresses several important questions regarding CWH. The authors aptly point out to the “gold-standard” of HIV cure strategies involve ATI. Placed in the context of ATI in CWH, it is invaluable to understand intact HIV DNA dynamics in this population. The manuscript was well-written and thoughtful. Some specific comments follow, but most fall under the modeling.

**Part II – Major Issues: Key Experiments Required for Acceptance**

Reviewer #1: 1. The definitions for “suppressed” and “viremic” participants are somewhat unclear. While I understand the post-stratification analysis presented in Figure S1, I believe it would be more appropriate to define viral suppression using a more stringent criterion, such as stratifying participants based on the area under the curve (AUC) of viral loads. Alternatively, how would the results change if a stricter definition of suppression were applied (e.g., <2 blips total, with viral loads <400 copies/mL)?

2. In several sections, the paper reads like a statistical analysis, but the biological implications and clinical relevance of the findings should be emphasized more. For instance, it is unclear why the authors compare “viral load decay from months 2-12 of ART” with “intact HIV DNA decay during year 0-1 of ART.” I recommend focusing on a smaller number of more meaningful comparisons. Additionally, correlations that did not reach statistical significance (e.g., Figure S6) should likely be removed, as they detract from the key findings.

3. The analysis of cytokine correlations, particularly in Figure S7, may not be highly relevant to the core focus of the study. These correlations seem to dilute the more compelling associations observed between Granzyme B (GzB) and faster viral load decay, and between IL-15 and the faster decay of intact proviruses. Since these associations did not survive multiple comparison corrections, their inclusion may need reconsideration.

4. The rationale for comparing the dynamics of intact and defective proviral DNA with TCR clonal dynamics (Figure 3F) is unclear. Given that clonal proliferation and contraction essentially balance out (i.e., homeostasis), and the observed decay in intact DNA (as reported previously), it is not surprising that these phenomena are largely independent. The statement that “defective DNA kinetics resemble CD4 count” (line 237) is not very conclusive and may be omitted.

5. The modeling portion of the study presents interesting conclusions but could benefit from clearer presentation. While the main conclusions outlined in lines 278-281 are understandable, the data supporting these statements should be presented more clearly. Figures 4C and D are difficult to interpret, and statistical support for the authors’ claims is needed. Was the modeling analysis necessary to demonstrate that short-term viremia does not permanently reseed the reservoir? Could this conclusion be drawn more simply by analyzing the longitudinal data already presented in the study?

Reviewer #2: see above

Reviewer #3: None.

**Part III – Minor Issues: Editorial and Data Presentation Modifications**

Reviewer #1: 1. It would be helpful to place the results in context with other studies on the dynamics of HIV reservoir markers in the early years of life: Veldsman et al. AIDS 2018; Katussime et al. J Virol 2020; Massanella et al. Clin Infect Dis 2021. This would provide a clearer understanding of how the findings contribute to the broader field.

2. Line 84: “CHW” should be corrected to “CWH.”

3. Line 221: “Repertoire sizes ranged from 4,904 to 61,288 sequences” – Is this referring to unique clonotypes or the total number of TCR sequences obtained?

4. Throughout the manuscript, the authors report HIV DNA levels per million T cells (i.e., CD4 and CD8). This is an unconventional method, and it would be helpful to clarify whether these measures were performed in PBMCs, T cells, or specifically in CD4+ T cells.

5. In Figure S7 and the associated correlation matrices, it would be useful to include p-values to provide transparency on the statistical significance of the reported associations.

6. The manuscript refers to “non-naïve cells.” It would be helpful to clarify whether this refers to CD45RA- or CD45RO+ cells, to ensure accuracy.

7. Granzyme B is not technically a cytokine, but rather an effector molecule or cytotoxin. This distinction should be clarified.

8. Line 345 mentions that the cells used for TCR sequencing were “resting” non-naïve T cells, but this contradicts the description in the methods section. The authors should reconcile these two statements to ensure consistency.

Reviewer #2: see above

Reviewer #3: 1. While the authors offer evidence of what subtypes are predominantly found in study population, is there any sequenced based evidence to confirm that this is consistent. This would also ultimately support the use of CS-IPDA within the tested subtypes.

2. Do the authors have plasma RNA (or DNA) sequence data from the suppressed subset where viremic episodes occurred, or in the viremic subset, to compare how and at what level, ongoing rounds of viral replication occurred? It would be interesting to consider the diversification of the virus since presumably the virus in plasma before ATI is what is observed in rebound viremia.

3. The modeling methods are quite nice. In parts a bit of an oversimplification but it appears that the authors intended this to enjoy some relaxed fits within modeling the data. Some specific points:

3A. Quite pedantic, but please define what “reservoir” and “latent reservoir” refer to in your study. I find by doing so lessens any squabbles. One can easily argue that the “reservoir” is that only of intact replication-competent proviruses. And by “latent”, referring to transcriptionally-silent or just all infected cells in the resting memory T cell phenotype? Personally, defective proviruses fall outside of the reservoir for this Reviewer, but in the context of the model and how the authors incorporate it, it is appropriate.

3B. Lines 529-548. Perhaps it can be supplemental, but the phrase “guesses” are interesting to use here. Did the authors use any a priori? How were the different models assessed—through progressive complexity and evaluated by AIC etc.? Why were the SDs assumed to be 1/10 mean?

3B.1. In general, the many assumptions and “guesses” could leave uninformed readers with some skepticism as to how the models were built and utilized. And ideas as the “passenger” hypothesis which (currently) does not include a reference that a reader could go back to, is something that needs to be defined.

3B.2. An overall suggestion is to use simplified jargon to support the decisions would be highly valuable and add definitions were appropriate.

PLOS authors have the option to publish the peer review history of their article (what does this mean?). If published, this will include your full peer review and any attached files.

Reviewer #1: No

Reviewer #2: **Yes: **Deborah Persaud, MD

Reviewer #3: No

**Figure resubmission:**
---

## [Editor Report · Decision Letter 1]

21 Feb 2025

Dear Dr. Reeves,

We are pleased to inform you that your manuscript 'Intact HIV DNA decays in children with and without complete viral load suppression' has been provisionally accepted for publication in PLOS Pathogens.

Best regards,

Mary F Kearney

Academic Editor

PLOS Pathogens

Susan Ross

Section Editor

PLOS Pathogens

Sumita Bhaduri-McIntosh

Editor-in-Chief

PLOS Pathogens

orcid.org/0000-0003-2946-9497

Michael Malim

Editor-in-Chief

PLOS Pathogens

orcid.org/0000-0002-7699-2064
---

## [Editor Report · Acceptance letter]

Dear Dr Reeves,

We are delighted to inform you that your manuscript, "Intact HIV DNA decays in children with and without complete viral load suppression," has been formally accepted for publication in PLOS Pathogens.

Best regards,

Sumita Bhaduri-McIntosh

Editor-in-Chief

PLOS Pathogens

orcid.org/0000-0003-2946-9497

Michael Malim

Editor-in-Chief

PLOS Pathogens

orcid.org/0000-0002-7699-2064